# Polymorphism in the Calpastatin Gene Alters Beef Tenderization in Excitable Cattle: A Preliminary Study

**DOI:** 10.3390/ani15111568

**Published:** 2025-05-27

**Authors:** Ana Cláudia da Silva, Patricia Maloso Ramos, Aline Silva Mello César, João Pedro Sousa do Vale, Saulo da Luz e Silva, Eduardo Francisquine Delgado

**Affiliations:** 1Departmento de Zootecnia, Escola Superior de Agricultura “Luiz de Queiroz”, Universidade de São Paulo, Piracicaba 13400-900, SP, Brazil; anaa.claaudia.silva@gmail.com (A.C.d.S.); dovalejps@gmail.com (J.P.S.d.V.); efdelgad@usp.br (E.F.D.); 2Departmento de Zootecnia, Faculdade de Zootecnia e Engenharia de Alimentos, Universidade de São Paulo, Pirassununga 13635-900, SP, Brazil; sauloluz@usp.br; 3Departmento de Ciência e Tecnologia dos Alimentos, Escola Superior de Agricultura “Luiz de Queiroz”, Universidade de São Paulo, Piracicaba 13400-900, SP, Brazil; alinecesar@usp.br

**Keywords:** beef, fragmentation, pH, proteolysis, stress response, tenderness

## Abstract

Tenderness in beef is of crucial importance not only to satisfy consumers but also to avoid the waste of such noble food, contributing positively to the sustainability of the production chain. *Bos taurus indicus* animals have great variability in beef tenderness and may naturally have a more excitable temperament. Excitable animals are difficult to handle and to transport, which could enhance pre-slaughter stress. Since calpastatin is a natural muscular inhibitor of proteolysis, and therefore tenderization, with corresponding activity upregulated by stress, tracking animals that are excitable and also harbor a calpastatin gene mutation related with reduced tenderness could positively contribute to addressing beef variability. In an experimental setting, cattle with divergent temperaments and harboring the specific mutation were identified, and beef quality was tested. Excitable cattle produced more variable and less tender beef compared to calm animals harboring the desirable allele combination. The observations herein could contribute to amplifying the basic knowledge of tenderization as well as indicate future research in commercial herds to contribute to breeding programs.

## 1. Introduction

The influence of biological mechanisms and their relationship with beef quality traits are understood by analyzing physical and biochemical processes that are involved in muscle-to-meat conversion. In this context, many studies have investigated factors affecting meat tenderization; some are related to animals, others to post mortem conditions. For example, cattle breed is well recognized to influence tenderization rate and extension, with an unfavorable condition related to *Bos taurus indicus* when compared to *Bos taurus taurus* beef [1]. Tenderness plays a central role in defining consumers’ expectations of beef [2]. Since consumers from diverse demographic origins have similar perceptions of sensory attributes of beef [3], and one failing attribute will significantly affect overall failure [4], tenderness must always be a consideration. Therefore, it is mandatory for industry to identify threshold acceptability and track it to deliver consistent beef tenderness.

Tenderness depends on changes in structural proteins in muscle. These changes begin with the sarcomere weaking, which results from protease activation. The calpain system is a calcium-dependent endogenous protease system responsible for myofibrillar proteolysis, which is the main system responsible for post mortem tenderization [5]. Calpastatin is an endogenous inhibitor of calpain that regulates in vivo myofibrillar protein degradation, and its gene (*CAST*) is intimately related to tenderness in beef [6] and determining the rate and extent of tenderization post mortem [7].

Animal temperament can influence beef tenderness [8]. Excitable animals are associated with greater occurrence of tough beef, which can be explained by the stimulation of the β-adrenergic receptor by catecholamines [9]. Its activation in the sarcolemma will initiate a cascade of intracellular events that culminate with calpastatin phosphorylation by protein kinase A [10]. Phosphorylation can enhance the affinity between calpastatin and calpains, which consequently reduces proteolysis [11].

Calpastatin is crucial for beef tenderization when the influence of tropical adapted cattle is relevant. Since *Bos taurus indicus* and its crosses govern most cow herds in Australia, the southern United States, and Brazil, comprising nearly one quarter of the beef produced in the world [12], calpastatin matters. Multiple studies have identified single nucleotide polymorphisms (SNPs) in the *CAST* gene as markers associated with tenderness in beef [13,14,15]. In this context, the occurrence of the G allele (GG and AG) of a *CAST* marker (position 97574679 on Btau4.0) in the *CAST* gene is associated with less tender beef than the AA genotype [16]. In Brazil, several studies have shown a significant association between *CAST* SNP markers and tenderness in Nellore beef [17]. Additionally, our research group previously investigated the association of temperament with calpastatin activity and degradation pattern post mortem [18,19]. To the best of our knowledge, the present study investigating the association between temperament and *CAST* genotypes in Nellore cattle is a novel one. 

Based on the provided evidence, the hypothesis that drove the present work is that excitable Nellore cattle harboring unfavorable SNPs in the *CAST* gene will have greater calpastatin activity and consequently produce less tender beef after aging. Therefore, the objective was to determine the association between *CAST* SNPs and tenderness in the *Longissimus* muscle of cattle with divergent temperaments.

## 2. Materials and Methods

### 2.1. Animals

All procedures involving the animals in this study were previously approved by the local Committee of Ethics in Animal Use (protocol number 6493190121).

Sixty Nellore animals were used in a nutrition experiment that was simultaneously conducted with the present study. The animals had an initial body weight of 350 ± 30 kg (average ± standard deviation) and an average age of 20 months and were obtained from the experimental herd of the University of Sao Paulo (Pirassununga, SP, Brazil). There were 30 immunocastrated (IM) and 30 non-castrated males (NC), from which a sub-group was selected after a temperament evaluation (see Section 2.3) and genotyping for the *CAST* gene (see Section 2.4). The animals were housed in a feedlot facility and fed for approximately 100 days.

### 2.2. Immunocastration

The immunocastrated animals (*n* = 30) received two doses of the immunocastration vaccine (Anti-GnRH; Bopriva^®^, Zoetis, São Paulo, Brazil) subcutaneously. The first dose was administered by a veterinarian 62 days before the beginning of the finishing period (at 18 months of age), and the second dose was administered when the animals were transferred from the pasture to the feedlot. According to the manufacturer’s instructions, a 60-day interval between doses results in a castration time of 120 days.

### 2.3. Animal Temperament Evaluation and Groups Formation

The temperament evaluation and first body weight determination were conducted during the first handling after the animals were transferred to the feedlot. The temperament was evaluated by the combination of two tests: (1) the chute score (CS) and (2) the exit velocity (EV). The first test was a visual and subjective evaluation in which a trained professional gave a score on a scale from 1 to 4 as soon as the animal entered the chute. On this scale, 1 = no movement, with head, ears, and tail relaxed; 2 = some movement, with the head and the ears upright; 3 = frequent but not vigorous movement and possible vocalization; and 4 = constant and strong movement, frequent vocalization, and showing great resistance (adapted from [20]). The second test was performed objectively by recording the time elapsed between the animal exiting the chute and the animal reaching a 5.40 m marked distance. The time was recorded by an observer, with the release of the chronometer as soon as the animal nose was observed leaving the chute and stopping the chronometer as soon as the rear touched the marked distance. The EV was reported in meters per second (m/s; adapted from [21]). The temperament index (TI) was calculated by numerically averaging both tests, as shown in the following formula: TI = (CS + EV)/2 [18].

Based on TI information, the overall average and standard deviation were calculated for all animals (*n* = 60), and a group of 23 animals was selected to represent two groups: calm (*n* = 11; TI ≤ 8.3) and excitable (*n* = 12; and TI ≥ 9.7, up to 17.7). Care was taken to select animals that were offspring from a diversity of sires so that the sire effect could be avoided once groups were formed. Therefore, there were nine and seven sires in the groups of calm and excitable animals, respectively. The remaining 37 animals could be classified as having an intermediate temperament and therefore were not further considered in the study.

### 2.4. DNA Extraction and Genotyping

During the first handling, tail hair-containing bulbs were removed from all animals for DNA extraction and packed in an identified paper envelope. Five bulbs from each animal were cut (0.5 cm) and transferred to a microtube containing extraction solution (200 mM NaOH). The microtubes were heated to 95 °C for 10 min to rupture the cellular membranes and consequently disperse the DNA. The samples were centrifuged at 13,000 rpm for 10 s to separate the DNA from the organelles and proteins present in the cells. Neutralizing solution (200 mM HCl and 100 mM Tris-HCl, pH 8.5) was added, and aliquots were stored in a freezer (−20 °C). DNA integrity was verified by electrophoresis on 1% agarose gel.

To amplify the sequence for the detection of the SNP of the *CAST* gene, a conventional PCR technique was used, using primers previously fabricated and the restriction enzyme *Dde*I (Desulfovibrio desulfuricans I; NCIB 83120). Primer fabrication followed the description by Allais et al. [16]. The *CAST*/*Dde*I polymorphism (23 bases) was amplified from 25 nmols of the DNA fragment using the forward primer sequence 5′CTCACGTGTTCTTCAGTGTTCTG3′ and the reverse primer sequence 5′CAACCCAAAGAAACATCAAACACAGT3′. The endonuclease restriction enzyme *Dde*I (Invitrogen, Waltham, MA, USA) used for digestion recognized and cleaved the 5′C^TNA_G3′ sequence located on chromosome 7. The amplification reactions had a final volume of 24 μL, containing 3 μL of DNA sample, 10 μL of PCR mix (PCRBIO Taq DNA Polymerase, PCR Biosystems Ltd., London, UK), 10 μL of ultrapure water, and 1 μL of the *CAST* primer, in a Bio-Rad C-1000 thermal cycler (Bio-Rad Laboratories, Hercules, CA, USA) set to denature initially at 95 °C for 3 min, followed by amplification for 30 cycles at 95 °C for 30 s, 1 min at 58 °C, and 1 min at 72 °C, with a final extension of 5 min at 72 °C. The PCR products were separated by electrophoresis on a 1.5% agarose gel.

The PCR product (5 μL) was digested with the restriction endonuclease *Dde*I (0.2 μL) in a solution that also contained 1.2 μL of digestion buffer and 2.6 μL of ultrapure water. The digestion was performed in a thermal cycler at 37 °C for 3 h. The DNA fragments were separated by electrophoresis via a 3.5% low-melting-temperature agarose gel using a molecular marker of 50 bp. The gel run was 40 min at 90 V, and the gel image was captured by a scanner under ultraviolet transillumination using Image Lab™ Molecular Imager Gel Doc™ XR Software version 1.0 (Bio-Rad Laboratories, Hercules, CA, USA). From the images of all the samples, it was possible to identify three genotypes: AA, AG, and GG (Figure 1). However, within the group previously selected by temperament (*n* = 23), only two animals were GG (one calm and one excitable). Both animals were removed from the final group of interest for the present study, which had only the AA and AG genotypes (*n* = 21; calm = 10, 5 AA and 5 AG; excitable = 11, 4 AA and 7 AG).

### 2.5. Slaughter and Sampling

The endpoint of the animals was previously set as a minimum of 5 mm of backfat thickness measured by ultrasound in the *Longissimus thoracis et lumborum* (LTL) muscle between the 12th and 13th ribs. Therefore, the animal was slaughtered as soon as it reached this endpoint. The slaughters were conducted in an experimental slaughterhouse belonging to the University of Sao Paulo (campus Pirassununga, SP, Brazil) and operating under the state inspection services (SISP). Due to the technical limitations of the facility, a maximum of 16 animals had to be slaughtered per day, following a previous classification based on the backfat thickness. Therefore, there were multiple dates of slaughter for the animals in the present study, which was accounted for in the statistical analysis. The animals were all slaughtered within a month.

The animals were transported in the morning of the same day by truck to the slaughterhouse, which is located 400 m from the feedlot. The animals fasted for 12 h prior to transport and always had free access to water.

#### 2.5.1. Plasma Lactate

At the time of exsanguination, blood was collected and transferred to a tube containing sodium fluoride and ethylenediaminetetraacetic acid (EDTA). The identified tubes were kept in an ice box until the end of the slaughter. The tubes with the blood were centrifuged at 2000× *g* for 20 min, and the plasma was transferred to a microtube and stored in a freezer (−20 °C). On the day of the analysis, the samples were transported to the Animal Metabolism Laboratory (located in Piracicaba, SP, Brazil), thawed, and subjected to analysis with a commercial kit (Lactate Enzymatic—Labtest Diagnostica S.A., Vista Alegre, MG, Brazil), after which the concentrations were determined by endpoint spectrophotometry. The results are reported in mg/dL.

#### 2.5.2. pH and Temperature Decline

The pH meter was calibrated prior to each measurement, and buffers were always maintained in the same room as carcasses. Calibration was performed using buffers 4 and 7, and buffer 7 also served as an indicator of accuracy, with a deviation of ±0.05 considered the threshold for repeated calibration. The pH and temperature decline were measured directly in the LTL muscle by the insertion of a portable pH meter (Hanna, model HI99163, Hanna Instruments, Barueri, Brazil) during the first 24 h post mortem (1, 3, 6, 9, 12, and 24 h), which was performed simultaneously with the temperature measurement by the insertion of the thermometer. The pH meter probe was always inserted in a new position near the anterior while avoiding damaged tissue and always in the same half of the carcass from each animal.

#### 2.5.3. Muscle Samples

At 1 h post mortem, small samples from the LTL were collected and immediately frozen in liquid nitrogen. At 24 h post mortem, the LTL was excised from the carcasses, fabricated into 2.5 and 1 cm steaks, vacuum packaged, and aged in a dark cold room (4 °C) for 1, 7, 14, and 28 days. After aging, the 2.5 cm steaks were used for instrumental color, cooking loss, and instrumental tenderness determinations, while a 1 cm steak was cut into small cubes, frozen with liquid nitrogen, and transferred to an ultra-freezer (−80 °C) for further analysis of myofibrillar fragmentation index.

### 2.6. Meat Quality Analysis

#### 2.6.1. Color

Beef color was measured in the LTL steaks after they were removed from the vacuum package and exposed to oxygen for blooming for 30 min under cold conditions (4 °C). The objective color was assessed using a CM2500d spectrophotometer (Konica Minolta Sensing Inc., Osaka, Japan), with illuminant A, an observer angle of 10°, and an 8 mm shutter opening, which was previously calibrated [22]. Three shots from different anatomical positions on the steak surface were taken and averaged to represent lightness (*L**), redness (*a**), and yellowness (*b**).

#### 2.6.2. Cooking Loss and Warner–Bratzler Shear Force

After assessing color, steaks were weighed and cooked in an electric oven (Flecha de Ouro Ind. e Com. Ltd., model F130/L, São Paulo, Brazil) previously heated to 170 °C. The steaks, equipped with a thermometer, were cooked to reach an internal temperature of 40 °C and then turned and cooked until 71 °C [23]. After cooking, the steaks were allowed to cool and weighed again to calculate cooking loss (CL), which was calculated as the percentage of weight lost during cooking, following the formula CL = [((initial weight − final weight)/initial weight) × 100].

Steaks were then wrapped in plastic film and kept in the refrigerator (4 °C) until the next day. Six cylinders were taken from each steak parallel to the muscle fiber alignment. The cylinders were sheared using a Warner–Bratzler blade (200 mm/min), and the average peak force used for shear was used to represent each steak, expressed in Newtons (N).

#### 2.6.3. Myofibrillar Fragmentation Index

The additional samples from the 1 cm LTL steak previously frozen were used for myofibrillar fragmentation index (MFI) determination, with modifications from Culler et al. [24]. All samples were assessed in duplicate. For extraction, 1 g of muscle sample was mixed with 10 mL of cold phosphate buffer (10 mM KH_2_PO_4_, 10 mM K_2_HPO_4_, 1 mM EDTA, 1 mM MgCl_2_ · 6H_2_O, 100 mM KCl, pH 7 at 4 °C), homogenized three times for 30 s with 30 s of rest on ice, and centrifuged at 1000× *g* for 15 min at 4 °C. After centrifugation, the supernatant was discarded, and a new buffer (10 mL) was added to the myofibrillar sediment, which was subsequently resuspended with a glass rod. This process was repeated three times. After the third time, the resuspension was performed with 5 mL of buffer, and the homogenate was filtrated to obtain the final concentrated solution. A protein assay using the biuret method was conducted, which allowed for the preparation of duplicate samples at a protein concentration of 0.5 mg/mL for absorbance determination using a spectrophotometer (Unico, model 1205, United Products & Instruments, Inc., Dayton, NJ, USA) at 540 nm. The four readings per sample were averaged, and the final value was multiplied by 200 and expressed as an index.

#### 2.6.4. Caseinolytic Inhibitory Calpastatin Activity

The total caseinolytic inhibitory activity of calpastatin was determined by the method proposed by Koohmaraie et al. [25], with some modifications. All steps were conducted under cold conditions unless stated otherwise. Samples from LTL 24 h post mortem (8 g) were weighed for extraction using extraction buffer (50 mM Tris, 10 mM EDTA at pH 8.3) supplemented with protease inhibitors (100 mg/L trypsin inhibitor, 2 mM PMSF, and 6 mg/L leupeptin). The homogenates were centrifuged at 24,652× *g* for 180 min, and the supernatants were transferred to dialysis tubes and dialyzed for 18 h in buffer containing 40 mM Tris, 5 mM EDTA at pH 7.3 with 10 mM β-mercaptoethanol. After dialysis, the homogenates were loaded onto chromatography columns previously filled with DEAE Sephacel resin (GE17-0500-01, Sigma Aldrich, São Paulo, Brazil) and equilibrated in buffer (50 mM Tris, 10 mM EDTA at pH 8.3). Once all of the homogenate passed through the column, the column was washed (40 mM Tris, 25 mM NaCl at pH 7.5) before elution was performed with elution buffer (40 mM Tris, 200 mM NaCl at pH 7.5). The enzymatic assay was conducted for each eluted sample, previously heated (95 °C). The assay was conducted by adding previously semi-purified m-calpain (from bovine lung), casein (Hammersten, E0789, Sigma Aldrich, São Paulo, Brazil), and calcium chloride (100 mM), and the activity was calculated based on changes in the absorbance readings at 278 nm (Shimadzu 1800, Shimadzu Corporation, Kyoto, Japan). A unit change in absorbance was considered proportional to a unit change in the inhibitory enzymatic activity of calpastatin, which is expressed as U/g of tissue.

### 2.7. Statistical Analysis

The experimental design was a randomized block (sexual category) in a factorial arrangement of 2 (temperament: calm and excitable) × 2 (genotype: AA and AG). The model was tested using the SAS statistical software OnDemand version 9.4 (SAS Inst. Inc., Cary, NC, USA) using a mixed model in which the fixed effects of block, temperament, genotype, and interaction were tested, as well as time post mortem, when applied, using slaughter and animal as random effects. The color, shear force, cooking loss, and myofibrillar fragmentation index were analyzed, with time considered as repeated measurements and covariance matrix adjusted as needed. Shear force data for beef aged 28 days were further investigated using an orthogonal contrast. For pH and temperature decline, statistical analysis for a spline’s regression fit was conducted using temperament/genotype and time post mortem or temperature in the model, as previously recommended by Hopkins et al. [26].

## 3. Results

### 3.1. Genotypic Frequencies

The genotypic frequencies obtained from the sixty animals showed a 50% occurrence of heterozygosity (AG), followed by 38.33% of favorable homozygosity (AA) and 11.67% of unfavorable homozygosity (GG; Figure 2).

### 3.2. Plasma Lactate Concentration

The plasma lactate concentration was not influenced by any interaction or investigated factor (Figure 3).

### 3.3. Muscle pH and Temperature Decline

The fitted lines based on the spline’s regression fit for temperature versus time post mortem (Figure 4A) indicated similarities between the groups formed by temperament and genotype, and the temperature measured in the LTL decreased from 39.0 ± 0.42 at 1 h to 5.3 ± 0.22 °C at 24 h post mortem. Similarly, the lines based on the spline’s regression fit for pH versus time post mortem showed similar patterns (Figure 4B), and it was noted that the green line (excitable AA group) was at the upper end. All pH values observed at 24 h post mortem were within the ‘normal’ range for beef (≤5.6).

The fitted lines based on the spline regression fit for pH by temperature were analyzed, and the dashed line representing the ideal pH/temperature window proposed by the Meat Standard Australia (MSA) was included in the figure. It was possible to observe more clearly that excitable animals harboring the AA *CAST* genotype showed a distinct pattern from that of the other three groups (Figure 4C). Although the lines were similar, the excitable AA group missed the window, while the other three groups were adjacent to the window at the cold end.

### 3.4. Beef Color

No two- or three-factor interactions were detected between temperament, genotype, and time for lightness (*L**), redness (*a**), or yellowness (*b**). During aging, the color attribute values increased from 1 to 7 and from 7 to 14, with no changes from 14 to 28 days (*p* < 0.001; Figure 5).

### 3.5. Cooking Loss, WBSF, and MFI

No interactions were detected between temperament, genotype, and time (triple or doubles) for cooking loss, which changed over time (*p* = 0.10), as expected. Losses were lower in steaks aged 1 day (28.1 ± 0.78%) and greatest in steaks aged 28 days (31.2 ± 0.97%).

No interactions were detected between temperament, genotype, and time (triple or doubles) for the WBSF, which changed (*p* < 0.001) over the aging period (Figure 6). The WBSF values steadily decreased from the first day to 7 days, from 7 to 14 days, and from 14 to 28 days of aging. No differences were observed for temperament or genotype alone.

Applying a threshold of 42.87 N for beef tenderness classification [27] showed that the only group that reached the threshold was the group of calm animals with the AA *CAST* genotype. Beef aged for 28 days from each group revealed that 80% of the calm AA group was classified as tender, while 40%, 25%, and 14.3% were classified as tender in the calm AG, excitable AA, and excitable AG groups, respectively. Further investigation using WBSF obtained at 28 days showed that the contrast between calm AA and all other animals was significant, with the former group showing lower values and, therefore, the most tender beef (*p* = 0.05; Figure 7).

A temperament versus time post mortem interaction was observed for MFI, with initial values (1 d) greater for beef from the excitable animals (*p* = 0.02; Figure 8). As expected, the index was influenced by the aging time (*p* < 0.001), with MFI showing the lowest values at 1 d, the greatest values at 28 days, and intermediate and similar values for beef aged 7 and 14 days.

### 3.6. Calpastatin Activity

An interaction (*p* < 0.10) between temperament and genotype influenced total calpastatin inhibitory activity (Figure 9). Animals classified as excitable AA presented lower inhibitory calpastatin activity than did excitable AG animals (*p* = 0.05), while no differences were observed within calm animals, which exhibited intermediate values. Therefore, greater variability in calpastatin activity was associated with excitable group.

## 4. Discussion

The plasma lactate concentration was not influenced by any of the factors analyzed (Figure 3) in the present study, although it was hypothesized that excitable animals would have greater concentration. Similarly, no differences in lactate concentrations were found between Nellore cattle with excitable and adequate temperaments [28]. However, previous studies reported greater plasma lactate concentrations in excitable cattle [9,29]. An increase in plasma lactate concentrations in animals with excitable temperaments is expected due to a greater stress response in muscles, which stimulates glycolysis. Exposure to stressors causes catecholamine release, which results in a cascade of events, including glycogenolysis, mobilization of hepatic and muscular glycogen stores, and increases in blood glucose and lactate [30]. Therefore, greater glycogenolysis associated with acute adrenergic stress can result in a two- or threefold greater increase in glycolysis than in oxidative phosphorylation, resulting in the accumulation of pyruvate diverted to lactate [31,32]. In the present study, it is possible to assume that the proper handling of animals in the loading and unloading from the truck, as well as the proximity between the feedlot and the slaughter facility (~400 m), were determinants of reduced stress before slaughter and, therefore, the reason for the absence of differences in plasma lactate concentrations between the animals with divergent temperaments.

Despite similar plasma lactate values, the lines representing pH decline as a function of temperature revealed that the excitable AA group showed a resistance to the decline in pH that was slightly more pronounced than that of the other three groups. Observation of the pattern of the lines revealed that the pH decline was faster in the calm AA group, followed by the calm AG group, the excitable AG group, and the most resistant excitable AA group (Figure 4C). This result partially corroborates previous observations in Nellore animals who also demonstrated a greater pH 48 h post mortem in more excitable animals [33]. However, in the present study, the pH values observed for calm and excitable animals were within the range classified as ‘normal’ for beef, that is, between 5.3 and 5.8. On the other hand, *Bos taurus taurus* crossbred steers classified as calm, intermediate, or excitable showed similar pH values at 48 h post mortem [8]. The rate of pH decline during the conversion of muscle to meat reflects the intensity of post mortem metabolism. One fundamental aspect of the interference of pH before rigor mortis (in bovines, at approximately 9 and 12 h post mortem) between temperaments would be associated with a greater μ-calpain proteolytic activity at more acidic pH, especially due to the temperature observed in the LTL muscle at approximately 1 h post mortem [34], which would reflect reduced tenderization in excitable animals. In the present study, all the lines representing pH decline as a function of temperature demonstrated a pattern distinct from that expected for beef with a superior quality, as proposed by the MSA window [26]. Importantly, the animals in the present study were slaughtered based on their backfat thickness reaching a minimum of 5 mm, which resulted in a similar temperature decline. Although previous comparisons of calm, intermediate, and excitable crossbred steers also revealed no differences in carcass temperature decline [8], it must be considered that it can be difficult for producers to finish non-castrated *Bos taurus indicus* with similar backfat thicknesses.

Differences in the pattern of LTL pH decline post mortem could impact beef color. However, beef lightness (*L**) was influenced only by aging time. Previously, observations related to the greater occurrence of darker beef in more excitable cattle [35] were associated with a greater ultimate pH in beef, which would influence the water holding capacity and lightness readings. In the present study, although the first day of aging resulted in lower *L** values (Figure 5A), it was not enough to classify the samples as DFD, as all groups presented normal pH values at 24 h post mortem. As reported before, meat lightness can be influenced by pH when the ultimate value is greater than 5.8, which can be associated with DFD [36]. The results of the present study point to CIE *L** *a** and *b** values that corroborate the efficient endpoint used, as no beef presented a dark color. The absence of the relationship between excitable animals and dark-cutting occurrence in the present study can be explained in part by the same reason that the plasma lactate levels were similar between the groups: a reduced duration of stress during pre-slaughter handling. It is possible that since glycogen stores were not depleted, its concentration promoted the proper extension of pH decline, the absence of an ultimate pH value above the normal range, and the absence of DFD or any atypical dark beef, which would be expected in beef from excitable *Bos taurus indicus*. The muscular glycogen concentration is a determinant of the ultimate pH, mostly under anaerobic conditions, as is the post mortem situation [37].

Cooking loss was not influenced by temperament, genotype, or interactions; however, it was influenced by time post mortem, with greater losses in beef aged longer (28 days), as expected. Although the present result is in accordance with previous findings for beef from Nellore cattle, which showed no differences in cooking loss between meat from animals of different temperaments [28], in the literature, there is a reported association between greater cooking loss in beef from excitable animals [38]. The reduced ability to hold water within the beef structure can influence juiciness and reduce global acceptance by consumers [39].

Tenderization was mainly affected by aging, as expected. Beef from all groups became tender as time progressed from 0 up to 28 days of aging (Figure 6). However, applying a tenderness threshold revealed that groups had distinct proportions of beef classified as tender, which was further explored. The ability of calm AA animals to produce tender beef after aging contrasts with the detrimental effect observed in WBSF from beef of calm animals harboring the unfavorable AG allele in the *CAST* gene or the beef of animals classified as excitable (Figure 7). These results are partially consistent with previous investigations showing an association between animal temperament and shear force, indicating a reduction in tenderness or a slower tenderization process in beef from excitable cattle [18,38]. The combined negative effect of reduced tenderness in the excitable and AG animals is potentially related to greater stimulation by the catecholamines of the agonist β-adrenergic receptors located in the sarcolemma [9,40]. The stimulation starts a cascade of intracellular events, including an increase in cAMP, which increases protein kinase A activity, responsible for the phosphorylation of calpastatin variants, whose profile is defined by genotype and enhances inhibitory activity, decreasing proteolysis. While excitable temperament seems to have a persistent negative impact on tenderization itself, the AG allele also had an impact on the rate and extent of tenderization in beef from calm animals. These results are in accordance with previous reports, in which significant additive effects of the *CAST*-2 G allele on WBSF and tenderness scores in beef of crossbred *Bos taurus taurus* and *Bos taurus indicus* were observed [41]. Studying *CAST* gene polymorphisms, it was observed that the AA genotype was superior to the heterozygous AG genotype in terms of beef tenderness in Nellore (*Bos taurus indicus*) and Nellore × *Bos taurus taurus* crossbreeds [42]. In this context, the positive relationship between shear force and the levels of calpastatin mRNA and activity, which would be detrimental for beef tenderness, were also observed previously [19]. However, it should be considered that longer aging could contribute to improved beef tenderization of calm AG and excitable animals.

The MFI is an indicator of muscle proteolysis, resulting in the breakdown of proteins that weaken and, when homogenized, form small myofibrillar fragments. Alterations in MFI were not observed during aging between phenotypes (temperament × genotype). However, an impact on fragmentation was observed between the temperaments during the time post mortem (temperament × time; Figure 8). However, the impact was observed only at the initial (1 day) aging, with greater fragmentation in beef of excitable animals. Additional aging time did not influence further fragmentation, as MFI values were similar between temperaments from 7 to 14 days, increasing from 14 to 28 days of beef aging. Since there were changes in the tenderization process, as evidenced by the WBSF values, there were similar changes in the MFI during the aging period. Additionally, the greatest difference in MFI values was observed from 1 to 7 days of aging. This observation is consistent with the fact that 60% of the myofibrillar degradation in beef occurs during the first two days of aging [43], indicating that further progression of fragmentation in beef in the present study was mostly inhibited.

The total caseinolytic inhibitory activity of calpastatin was influenced by the interaction between temperament and genotype (Figure 9), with excitable AA animals showing lower activity than excitable AG animals, with a difference of 0.7 U/g of tissue. These results add to previous findings that investigated different SNPs of the *CAST* gene associated with tenderness, in which allele A was considered favorable and G was considered unfavorable [16]. The new component added in the present study, animal temperament, is a determining factor for the observed difference in *Bos taurus indicus* animals, corroborating the possibility that post-translational modifications in excitable cattle act on the expression profile of the *CAST* gene. The alteration identified in the inhibitory activity between genotypes, only for excitable animals, which had an impact on tenderization as gauged by WBSF, corroborates the relationship between temperament and inhibitory calpastatin activity in Nellore cattle [18]. Additionally, there seems to be a difference within temperament that is linked to calpastatin expression level and its degradation post mortem [19], which is unfavorable for excitable animals. These results point to greater expression of *CAST* in beef of excitable AG animals, with a composition of gene expression and alteration in activity dependent on the animal’s temperament, with a detrimental impact on the rate and extent of beef tenderization.

The results presented herein must be carefully considered, as this is a preliminary study with a reduced number of animals within each group, all belonging to the university herd. However, since the animal model (Nellore) is the most relevant breed in Brazil, it is important to present and discuss these preliminary data, as they can contribute to drive future investigations using commercial genetics and a larger number of animals.

## 5. Conclusions

This preliminary study suggests that excitable animals at slaughter may pose challenges for beef quality by exhibiting resistance to pH decline and increased variability in calpastatin activity. Tenderization is favorable in beef of calm AA allele animals at the *CAST* gene polymorphism (position 97574679 on Btau4), while animals harboring the unfavorable AG allele or showing excitable temperament are not prone to produce tender beef, at least after 28 days of aging. The findings partially confirm our initial hypothesis, highlighting that both temperament and *CAST* genotype influence post mortem proteolytic processes. Given the high frequency of the AG genotype observed herein, which potentially reflects that of commercial herds, and the potential deleterious impact of the GG genotype as well, further research should explore the molecular mechanisms by which temperament interacts with *CAST* gene expression and proteolysis regulation. These insights may provide critical opportunities to optimize breeding strategies and handling practices, aiming to improve tenderness consistency and add value to the Brazilian beef industry.

## Figures and Tables

**Figure 1 animals-15-01568-f001:**
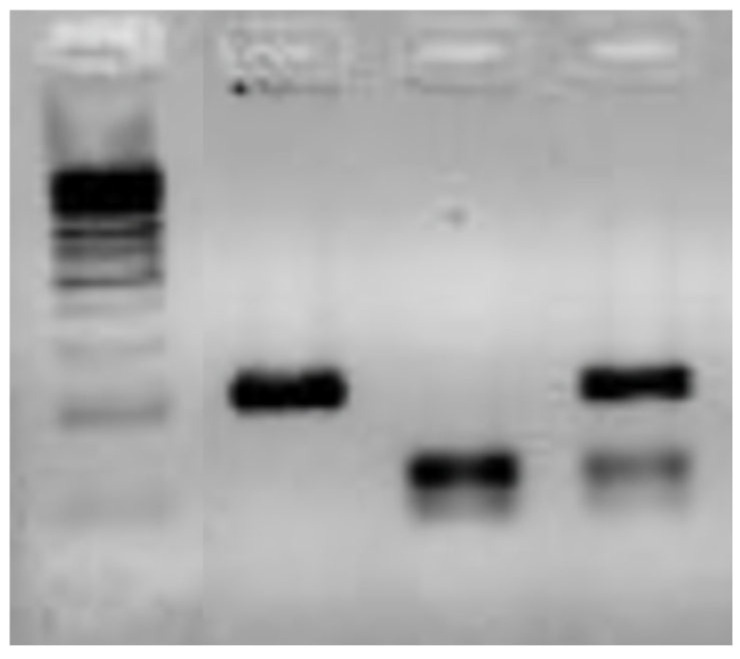
Agarose gel electrophoresis (3.5%) of products amplified by PCR and cleaved with the restriction endonuclease *Dde*I for the SNPs in the calpastatin gene. First lane = molecular marker of 50 bp; second lane = homozygous individual for allele G (GG); third lane = homozygous individual for allele A (AA); fourth lane = heterozygous individual (AG).

**Figure 2 animals-15-01568-f002:**
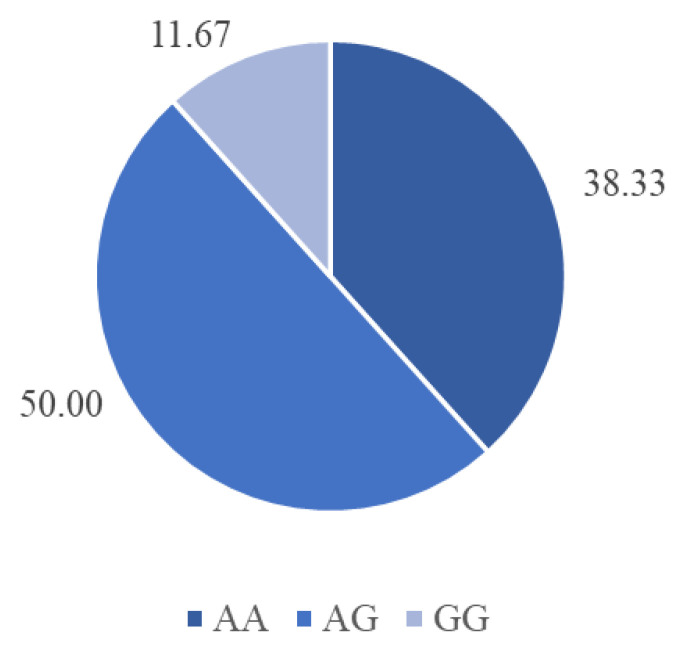
Genotypic frequencies (%) in the calpastatin gene (AA, AG, or GG) of Nellore cattle amplified by PCR and cleaved with the restriction endonuclease *Dde*I for the SNPs in a specific location (for more information, see Allais et al. [16]; (*n* = 60)).

**Figure 3 animals-15-01568-f003:**
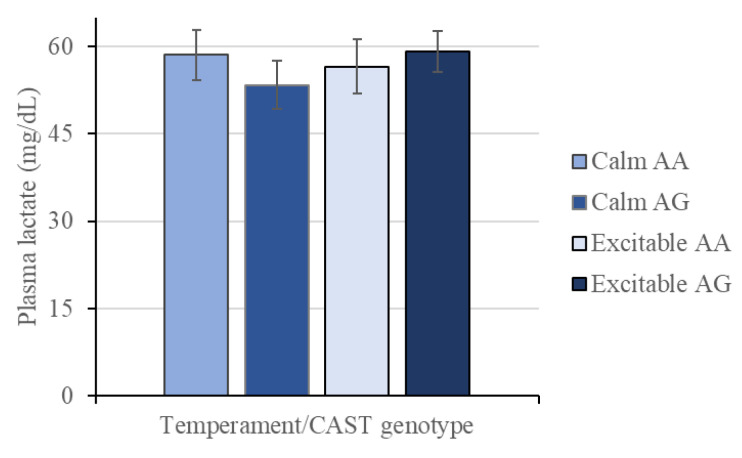
Plasma lactate (mg/dL) from Nellore cattle classified as calm or excitable and harboring a specific genotype in the calpastatin gene (AA or AG). Means ± standard errors.

**Figure 4 animals-15-01568-f004:**
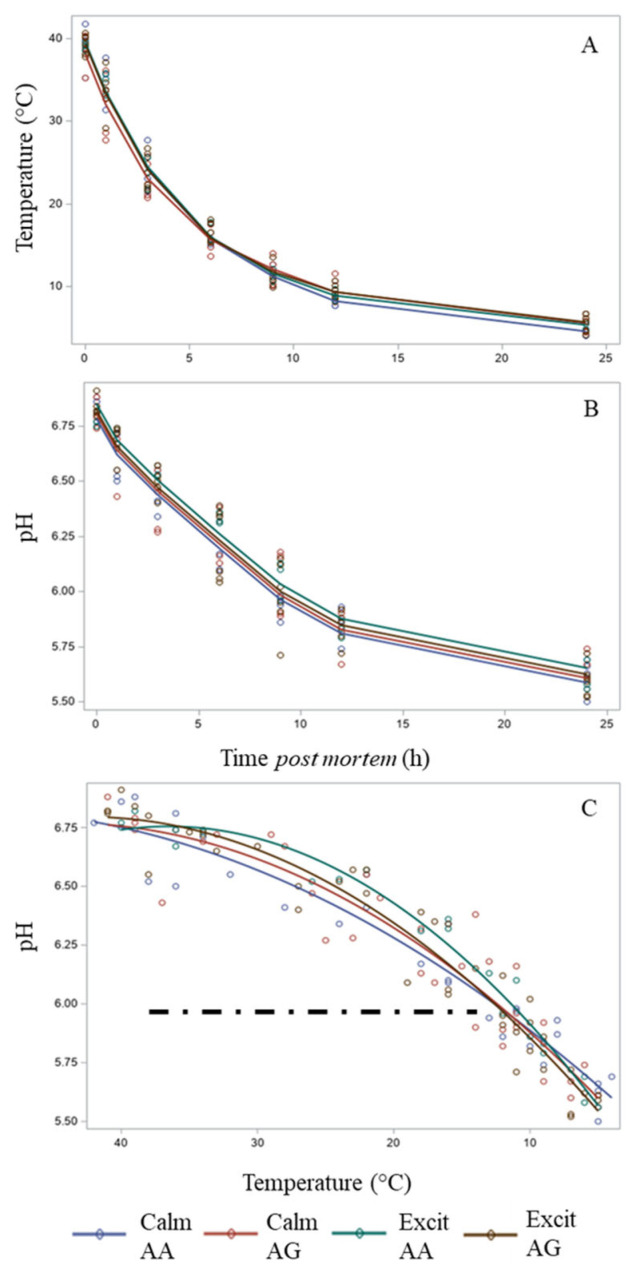
Lines of best fit based on the splines’ modeling of pH and time post mortem (**A**), temperature and time post mortem (**B**), and pH and temperature (**C**) for carcasses from Nellore cattle classified as calm or excitable, harboring a specific genotype in the calpastatin gene (AA or AG), with the ideal pH/temperature window shown as the dashed black line (pH/temperature window defined as temperature at pH 6 in the *Longissimus* < 35 °C and >12 °C).

**Figure 5 animals-15-01568-f005:**
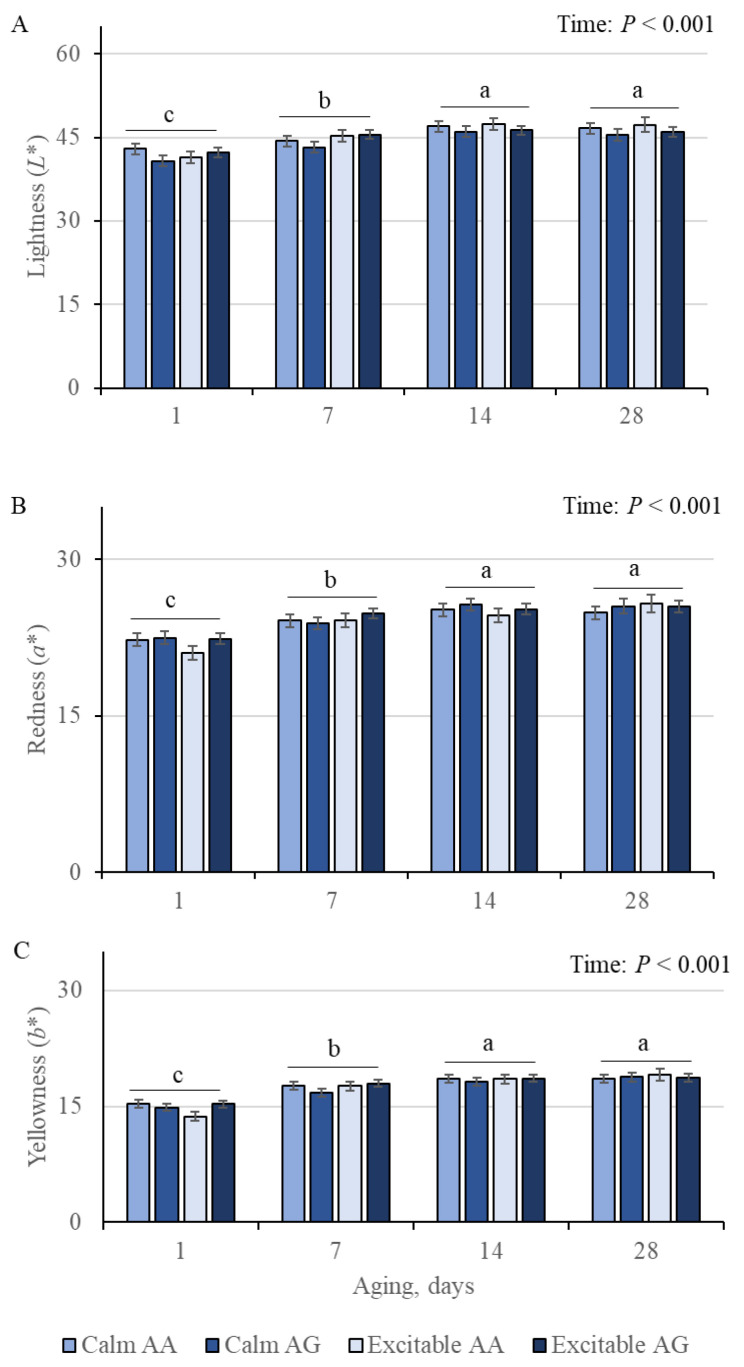
Beef lightness (**A**), redness (**B**), and yellowness (**C**) from Nellore cattle classified as calm or excitable, harboring a specific genotype in the calpastatin gene (AA or AG), and aged up to 28 days. ^a–c^ Indicates differences (*p* < 0.05) among aging times. Means ± standard errors.

**Figure 6 animals-15-01568-f006:**
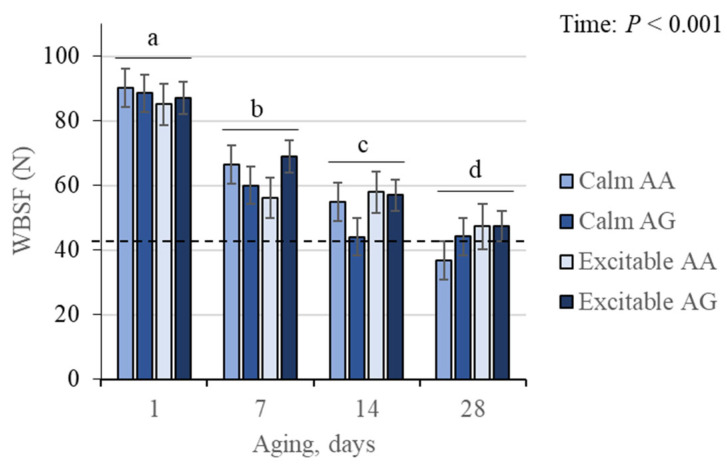
Warner–Bratzler shear force (WBSF) in beef from Nellore cattle classified as calm or excitable, harboring a specific genotype in the calpastatin gene (AA or AG), and aged up to 28 days. Dashed line represents a threshold for tenderness (42 N). ^a–d^ Indicates differences (*p* < 0.05) among aging times. Means ± standard errors.

**Figure 7 animals-15-01568-f007:**
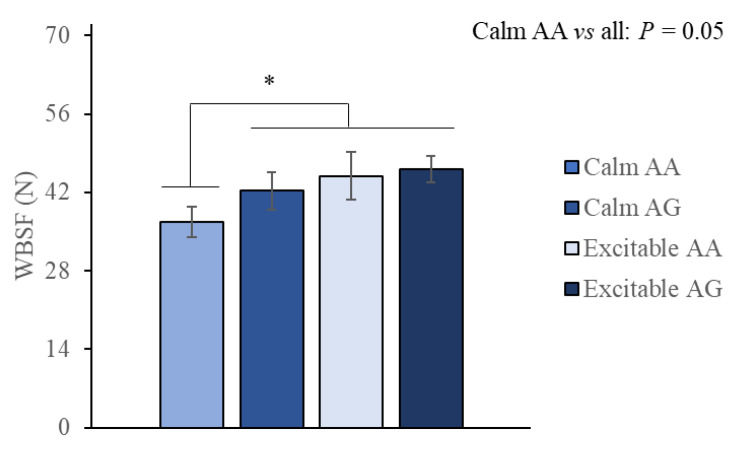
Warner–Bratzler shear force (WBSF) of beef from Nellore cattle classified as calm or excitable, harboring a specific genotype in the calpastatin gene (AA or AG), and aged for 28 days explored by an orthogonal contrast. * *p* = 0.05.

**Figure 8 animals-15-01568-f008:**
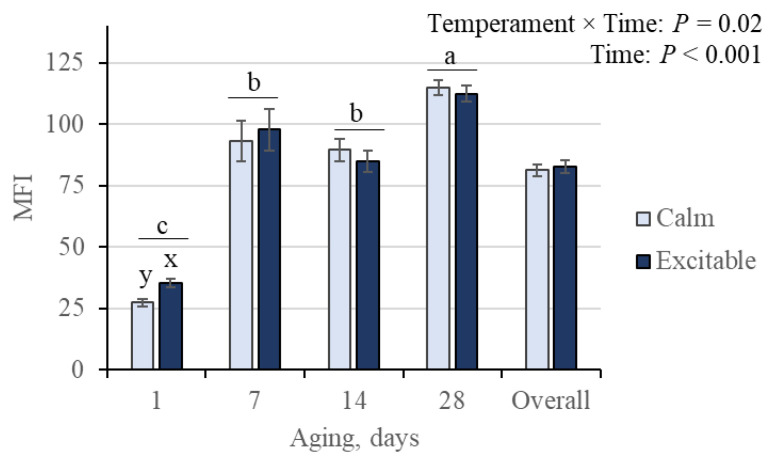
The myofibrillar fragmentation index (MFI) in beef from Nellore cattle classified as calm or excitable, harboring a specific genotype in the calpastatin gene (AA or AG), and aged up to 28 days. ^a–c^ Indicates differences (*p* < 0.05) among aging times; ^x,y^ indicates differences (*p* < 0.05) between temperaments within aging. Means ± standard errors.

**Figure 9 animals-15-01568-f009:**
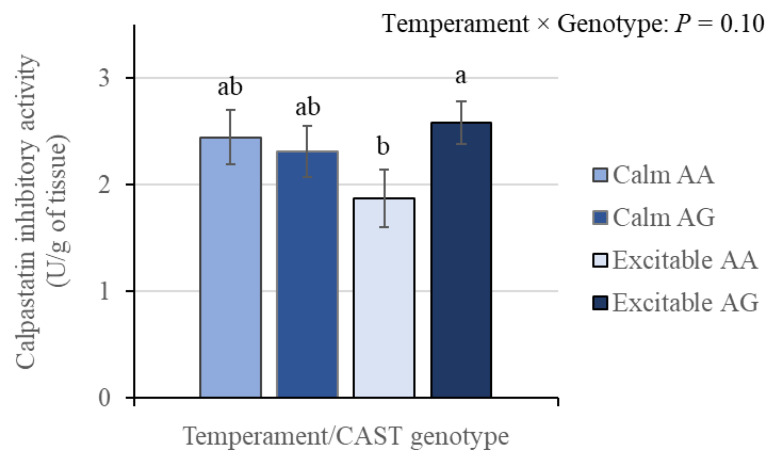
Calpastatin inhibitory activity (U/g of tissue) in beef from Nellore cattle classified as calm or excitable and harboring a specific genotype of the calpastatin gene (AA or AG). ^a,b^ Indicates differences (*p* < 0.05) between groups. Means ± standard errors.

## Data Availability

The raw data supporting the conclusions of this article will be made available by the authors upon request.

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
