# Peer review of "Polymorphism in the Calpastatin Gene Alters Beef Tenderization in Excitable Cattle: A Preliminary Study"

_animals, 2025, doi:10.3390/ani15111568_

Round 1

Reviewer 1 Report

Comments and Suggestions for Authors

Dear authors,

The study's objective was to evaluate the association between calpastatin (CAST) polymorphisms and tenderness in beef from Nellore cattle with divergent temperaments. In general, the manuscript has scientific merit and should be published in the Animals journal after major revision. General and specific suggestions and comments see below.

General comments:

  • Improve the abstract. It does not present the number of animals in the study. In addition, the abstract only presents, Rarionale, Objective, Methodology and Conclusion without Results.
  • Use genes abbreviations in italic.
  • How did you analyze or evaluate data considering both groups of the original trial (immunocastrated and non-castrated) animals? Since castration has an impact on tenderness.
  • Based on your results is it possible to conclude that the heterozygosity of AG slowed the tenderization process and extension in excitable animals? Explain better here and in the manuscript.

Specific comments:

L35: Aging. Check all manuscript.

L72: Remove Florida. In Texas and other States is also important.

L76: Complement the statement: “In Brazil, several studies have shown a significant association between CAST SNP markers and tenderness in Nellore beef [17].” How about temperaments? Are there studies to report? Or is your study a novelty?

L99: Bopriva or Bovipriva? Pfizer?

L162-163: How many animals were AA and AG?

L183: Why did you analyze blood lactate after slaughter?

L199: What is LL? Which part of the carcass did you collect muscle samples?

L332: How about Calm and Excited group. Was there any statistical difference in WBSF? How about AA and AG? You only reported no interaction effect.

L345: How many animals do you have classified as Calm AA?

L369: Important to point that Excitable AG does not have difference for calpastatin activity than Calm animals, regardless the allele.

Author Response

Dear reviewer,

We appreciate the comments and suggestions. We worked to improve our manuscript following your recommendations. Answers here are written in blue and modifications in the manuscript in red.

Comments and Suggestions for Authors

Dear authors,

The study's objective was to evaluate the association between calpastatin (CAST) polymorphisms and tenderness in beef from Nellore cattle with divergent temperaments. In general, the manuscript has scientific merit and should be published in the Animals journal after major revision. General and specific suggestions and comments see below.

We appreciate the acknowledgement.

General comments:

  • Improve the abstract. It does not present the number of animals in the study. In addition, the abstract only presents, Rarionale, Objective, Methodology and Conclusion without Results.

The abstract was modified to include the number of animals as well as more results.

  • Use genes abbreviations in italic.

It was modified accordingly.

  • How did you analyze or evaluate data considering both groups of the original trial (immunocastrated and non-castrated) animals? Since castration has an impact on tenderness.

Both groups (immunocastrated and non-castrated) were represented in the selected group for the study.

  • Based on your results is it possible to conclude that the heterozygosity of AG slowed the tenderization process and extension in excitable animals? Explain better here and in the manuscript.

Based on the frequency of beef classified as tender, as well as the occurrence of beef below the tenderness threshold (Figures 6 and 7), as well as the greater calpastatin activity in excitable AG which also is more variable within excitable animals, we understand that yes, it is possible to conclude that heterozygosity associated with excitable temperament is detrimental to beef tenderization. We updated the conclusion to better represent this rationale.

 Specific comments:

L35: Aging. Check all manuscript.

It was modified throughout the manuscript.

L72: Remove Florida. In Texas and other States is also important.

Ok, it was deleted.

L76: Complement the statement: “In Brazil, several studies have shown a significant association between CAST SNP markers and tenderness in Nellore beef [17].” How about temperaments? Are there studies to report? Or is your study a novelty?

There are other studies from our research group that previously investigated the relationship between temperament and tenderness. However, the association between temperament and CAST genotypes in Nellore cattle is a novelty. We updated the sentence accordingly.   

L99: Bopriva or Bovipriva? Pfizer?

Bopriva, Zoetis. We updated, thanks.

L162-163: How many animals were AA and AG?

9 animals were AA (5 calm and 4 excitable) and 12 animals were AG (5 calm and 7 excitable). The information was added to the manuscript.

L183: Why did you analyze blood lactate after slaughter?

We analyzed blood lactate after the slaughter based on reports that showed strong relationship between lactate and stress immediately prior to slaughter. Our intention was to verify the relationship between animals previously classified as excitable and lactate levels. However, the stress prior to slaughter, in the case of present study, was not enough to exacerbate differences. 

L199: What is LL? Which part of the carcass did you collect muscle samples?

LL is Longissimus lumborum muscle. However, the samples were collected at the end of thoracic portion and beginning of lumbar portion. Therefore, we decided to modify the abbreviation to LTL, representing the Longissimus thoracis et lumborum muscle. It was modified throughout the manuscript.

L332: How about Calm and Excited group. Was there any statistical difference in WBSF? How about AA and AG? You only reported no interaction effect.

No differences were observed for the main effects either. The information was added to the manuscript.

L345: How many animals do you have classified as Calm AA?

The information was added previously (MM section). In this group, there are 5 animals.

L369: Important to point that Excitable AG does not have difference for calpastatin activity than Calm animals, regardless the allele.

Yes, we reinforced that notion in the manuscript. Thanks.

Reviewer 2 Report

Comments and Suggestions for Authors

Dear Editor and Authors;

The authors of the present study have conducted a comprehensive analysis of the impact of calpastatin gene polymorphisms on the tenderness of beef in Nellore cattle, taking into account the animals' temperament. The introduction offers a comprehensive overview of the subject and explicitly articulates the study's hypothesis and aims. The Materials and Methods part is organized, comprehensive, and replicable. The results are effectively represented; nevertheless, the use of colorful figures would have enhanced their impact. The Discussion section compares the results with analogous studies and explores potential explanations. The Conclusion section succinctly and explicitly articulates the study's conclusions. I contend that the work warrants acceptance following the resolution of the minor flaws outlined below.

Best regards.

Comments:

L90: Please specify whether the live weights are presented as SD or SEM.

L176-178: Please also specify how many days it took to complete the slaughter process.

L215-221: Please specify how many technical replications were used for the color measurements.

L251: Please use “Koohmaraie et al. [24]” instead of “[24]”.

L295, L324, L338, L350: Having the figures in this section in color would be beneficial for visual appeal.

Author Response

Dear reviewer,

We appreciate the comments and suggestions. We worked to improve our manuscript following your recommendations. Answers here are written in blue and modifications in the manuscript in red.

Comments and Suggestions for Authors

Dear Editor and Authors;

The authors of the present study have conducted a comprehensive analysis of the impact of calpastatin gene polymorphisms on the tenderness of beef in Nellore cattle, taking into account the animals' temperament. The introduction offers a comprehensive overview of the subject and explicitly articulates the study's hypothesis and aims. The Materials and Methods part is organized, comprehensive, and replicable. The results are effectively represented; nevertheless, the use of colorful figures would have enhanced their impact. The Discussion section compares the results with analogous studies and explores potential explanations. The Conclusion section succinctly and explicitly articulates the study's conclusions. I contend that the work warrants acceptance following the resolution of the minor flaws outlined below.

Best regards.

We appreciate it. Thank you.

Comments:

L90: Please specify whether the live weights are presented as SD or SEM.

The live weight is presented as average followed by standard deviation (SD). The information was added to manuscript.

L176-178: Please also specify how many days it took to complete the slaughter process.

We added the information to the manuscript, but the animals were all slaughtered within a month.

L215-221: Please specify how many technical replications were used for the color measurements.

The color was measured in one steak per animal/per aging period. Therefore, if there was 5 animals in the group, 5 steaks (one of each animal) was assessed at three different anatomical positions (3 shots). The average of these 3 measures represented the steak for that specific animal.

L251: Please use “Koohmaraie et al. [24]” instead of “[24]”.

We corrected. Thanks.

L295, L324, L338, L350: Having the figures in this section in color would be beneficial for visual appeal.

We updated all figures in color, thanks.

Reviewer 3 Report

Comments and Suggestions for Authors

The manuscript: “Polymorphism in the calpastatin gene alters beef tenderization in excitable cattle”, is research on beef tenderization that is undoubtedly important due to its relevance to consumers. Furthermore, Nellore cattle and their temperament play a crucial role in the global market.
However, some issues need to be addressed in the manuscript, which I will now outline for the authors.

Comments 1: Line 31: Please remove the word “type”. Single nucleotide polymorphism is a SNP.
Comments 2: Line 75-76: The introduction could be improved with more information about the SNP (position, effects, sense or missense, AA, etc). 
Comments 2: Please explain, if sixty Nellore animals were used in the nutrition experiments, why the number was reduced for SNP detection (n=23)? 
Comments 3: Line 140: The restriction enzyme must be in cursive DdeI, please correct it. 
Comments 4: Lines 142 and 143: The authors did not specifically the primer design used in the research. Please explain, do the primers correspond to Allais et al. (2011)? 
Comments 5: I suggest that the agarose gel image could be improved. 
Comments 6: Line 194-204: Please detail the number of measurements in the carcass for pH and temperature. 
Comments 7: Line 215-220: Please explain, why the color measurements were made from vacuum samples?.  And how many measurements were made?
Comments 8: Line 226-227: For a better understanding, please write the cooking loss formula. 
Comments 9: Previously to plasma lactate results, I think is very relevant to present allelic and genotypic frequencies of the calpastatin gene. 
Comments 10: In Figure 2, the authors present four classifications: calm AA, calm AG, excitable AA, and excitable AG. Please provide the number of individuals for each classification. This is highly relevant to the statistical analysis because, if the GG genotype was removed, AA and AG (n = 21; 10 calm and 11 excitable) would have only 5 individuals for analysis in each classification?
Comments 11: Please explain why the authors did not consider increasing the number of animals or genotyping the initial sixty Nellore cattle to identify the different genotypes and the effect of the G and A alleles on beef tenderness, rather than limiting the discussion solely to the negative impact of AG heterozygosity.

Author Response

Dear reviewer,

We appreciate the comments and suggestions and worked to improve our manuscript following your recommendations. Thank you. Here answers to each comment is written in blue and in the manuscript, in red.

Comments and Suggestions for Authors

The manuscript: “Polymorphism in the calpastatin gene alters beef tenderization in excitable cattle”, is research on beef tenderization that is undoubtedly important due to its relevance to consumers. Furthermore, Nellore cattle and their temperament play a crucial role in the global market.
However, some issues need to be addressed in the manuscript, which I will now outline for the authors.

Thank you for your contribution to our study.

Comments 1: Line 31: Please remove the word “type”. Single nucleotide polymorphism is a SNP.

Ok, deleted.

Comments 2: Line 75-76: The introduction could be improved with more information about the SNP (position, effects, sense or missense, AA, etc).

Thank you for the comment. However, it is our understanding that the introduction should provide enough information for the broad comprehension of the manuscript. Therefore, the mentioned information was shown (and kept) in the MM section.

Comments 2: Please explain, if sixty Nellore animals were used in the nutrition experiments, why the number was reduced for SNP detection (n=23)? 

We genotyped all the animals. The number was reduced from 60 to 23 based on their temperament tests, because our objective was to compare extremes in terms of temperament. Therefore, after an initial evaluation of temperament we selected a group of 11 animals classified as calm to compare with 12 animals classified as excitable. The remaining 37 were considered as intermediate (neither calm or excitable). Therefore, it is not interesting for the present study. After genotyping all the animals, among those previously selected based on temperament (n=23) two animals were GG and were removed from the group based on the absence of replication. Finally, 21 animals were the group in the study. We added a better explanation to the manuscript, regarding the removed animals.

Comments 3: Line 140: The restriction enzyme must be in cursive DdeI, please correct it.

We are not sure whether the reviewer is asking to write the full name of the restriction enzyme or to use italic in the abbreviation. Therefore, we added the complete name and reference (Desulfovibrio desulfuricans I; NCIB 83120), as well as italicized the first three letters (DdeI).

Comments 4: Lines 142 and 143: The authors did not specifically the primer design used in the research. Please explain, do the primers correspond to Allais et al. (2011)?

Yes, the primer design used corresponds to that described by Allais et al., 2011. The information was added to the manuscript.

Comments 5: I suggest that the agarose gel image could be improved.

 To the best of our knowledge, we work to improve the image.

Comments 6: Line 194-204: Please detail the number of measurements in the carcass for pH and temperature. 

The number of pH and temperature measurements corresponds exactly to the number of times post mortem measured for each carcass, as explained in the manuscript. In this case, the group mean and standard error of the mean is obtained by the number of carcasses measured. For example, if that group has 5 animals, the pH at 1 h post mortem for that group is from 5 carcasses.

Comments 7: Line 215-220: Please explain, why the color measurements were made from vacuum samples?.  And how many measurements were made?

As mentioned in the manuscript, the color was measured after the vacuum package was removed and the steak surface was allowed to bloom for 30 min under cold conditions. Steaks were vacuum packaged to age, but color was measured without the packaging. Three shots were taken from steak surface and averaged for represent that steak. This information is in the manuscript as well.

Comments 8: Line 226-227: For a better understanding, please write the cooking loss formula.

The formula was added to the manuscript.

Comments 9: Previously to plasma lactate results, I think is very relevant to present allelic and genotypic frequencies of the calpastatin gene. 

We included the genotypic frequency to the manuscript as requested.

Comments 10: In Figure 2, the authors present four classifications: calm AA, calm AG, excitable AA, and excitable AG. Please provide the number of individuals for each classification. This is highly relevant to the statistical analysis because, if the GG genotype was removed, AA and AG (n = 21; 10 calm and 11 excitable) would have only 5 individuals for analysis in each classification?

The information was added to the MM section.

Comments 11: Please explain why the authors did not consider increasing the number of animals or genotyping the initial sixty Nellore cattle to identify the different genotypes and the effect of the G and A alleles on beef tenderness, rather than limiting the discussion solely to the negative impact of AG heterozygosity.

We did genotype the sixty animals. However, the frequency of GG allele was low for the selection based in temperament (2 animals, 1 calm and 1 excitable), as explained in the manuscript. Therefore, we did not have enough replication to investigate the effect of G and A and decided to report AA and AG alleles in the study, since the statistical analysis showed that heterozygosity associated with temperament presented detrimental results in terms of calpastatin inhibition of calpain caseinolytic activity.

Round 2

Reviewer 3 Report

Comments and Suggestions for Authors

Dears Authors, 

Thank you for reviewing and correcting the suggestions in the manuscript. 

Although there are still inaccuracies in the manuscript, for example, the introduction states that 'the occurrence of alleles GG and AG at a specific position in the CAST gene is associated with less tender beef than the AA genotype' (lines 74–75)—the exact position should be specified, as multiple A/G SNPs have been described in this gene, which could lead to confusion.

However, the new information generated appears to be more relevant due to the low number of individuals, raising concerns about the validity of the statistical analysis. That said, the sample's representativeness remains very low with only 5 AA and 5 AG animals and an excitable group consisting of 11 animals (4 AA and 7 AG). Could more animals be included?

Author Response

Dears Authors, 

Thank you for reviewing and correcting the suggestions in the manuscript. 

Although there are still inaccuracies in the manuscript, for example, the introduction states that 'the occurrence of alleles GG and AG at a specific position in the CAST gene is associated with less tender beef than the AA genotype' (lines 74–75)—the exact position should be specified, as multiple A/G SNPs have been described in this gene, which could lead to confusion.

Thank you for the comment, we added the position as follow: “The G allele of a CAST marker (position 97574679 on Btau4.0)”.

However, the new information generated appears to be more relevant due to the low number of individuals, raising concerns about the validity of the statistical analysis. That said, the sample's representativeness remains very low with only 5 AA and 5 AG animals and an excitable group consisting of 11 animals (4 AA and 7 AG). Could more animals be included?

Unfortunately for the present study we cannot add more animals. However, we understand the concern and we added the preliminary characteristic of the study at the end of the discussion chapter to emphasize that results must be carefully considered.